# Paleogene India-Eurasia collision constrained by observed plate rotation

Xiaoyue Wu [1,2,3], Jiashun Hu [2] ✉, Ling Chen [1,3], Liang Liu[4] & Lijun Liu [1,5] ✉

The Cenozoic India-Eurasia collision has had profound impacts on shaping the Tibetan plateau, but its early history remains controversial due to uneven availability of constraints. Recent plate reconstructions reveal two prominent counterclockwise rotation (azimuthal change) rate peaks of the Indian plate at 52-44 and 33-20 Ma, respectively, which could bear key information about this collision history. Using fully dynamic three-dimensional numerical modeling, we show that the first rotation rate peak reflected the initial diachronous collision from the western-central to eastern Indian front, and the second peak reflected the full collision leading to strong coupling between India and Eurasia. Further comparison with observation suggests that the initial and complete India-Eurasia collision likely occurred at 55 ± 5 and 40 ± 5 Ma, respectively, an inference consistent with key geological observations. We suggest that this collision history is instructive for studying the tectonic history of the Tibetan plateau and its surrounding areas.

When India collided with Eurasia and how this process varied along the strike of the Indian front have been debated for decades[1,2]. Among the available constraints, the abrupt deceleration of the Indian plate's drifting rate at ca. 50 Ma is often considered a direct response to the initial collision[3–6]. However, other geological proxies such as stratigraphy, sedimentology, metamorphism, and paleomagnetism along the suture often provide inconsistent ages of the initial collision[2]. A popular alternative explanation is diachronous collision[7–12], but this model also bears large uncertainties with the proposed initial collision occurring in the west[7,8], the central[9–11], or even the east of the Indian front[12]. Better resolving the Paleogene collision history is critical for understanding the tectonic evolution of the Tibetan plateau and the associated geological and climatic implications, such as the plateau uplift[11] and the monsoon evolution[13].

The asymmetric distribution of landmass and seafloors within the Indian plate along the convergent boundary made the plate's Cenozoic kinematic history highly sensitive to forces associated with its collision with Eurasia (Fig. 1 and Supplementary Fig. 1). The enormous collision-induced resistive force could not only reduce the speed of plate convergence, but also generate substantial resistive torques that may cause additional rotation of the incoming plate. Specifically, different suturing scenarios could generate distinct resistive torques, which may result in different patterns and rates of the Indian plate rotation. Observationally, some early paleomagnetic studies noticed significant Cenozoic rotation motions of the Indian plate[1,14], but a continuous and quantitative description of this rotation history is lacking, so is the underlying geodynamic mechanism.

In this study, we analyze the Cenozoic rotation motion of the Indian plate based on multiple recent reconstructions[15–18]. We find two peaks in the rate of counterclockwise rotation during the Paleogene, which may bear new constraints on the process of the India-Eurasia collision. To quantitatively explore the mechanisms behind this rotation motion, we build fully dynamic three-dimensional subduction-collision models and test end-member scenarios of diachronous collision, including west-, middle- and east-bulging configurations of Greater India (the subducted northern extension of the Indian subcontinent)[19,20]. By comparing the observed rotation history with that predicted from the geodynamic models, we propose that the initial

[1]State Key Laboratory of Lithospheric Evolution, Institute of Geology and Geophysics, Chinese Academy of Sciences, 100029 Beijing, China. [2]Department of Earth and Space Sciences, Southern University of Science and Technology, 518055 Shenzhen, China. [3]College of Earth and Planetary Sciences, University of Chinese Academy of Sciences, 100049 Beijing, China. [4]State Key Laboratory of Isotope Geochemistry, Guangzhou Institute of Geochemistry, Chinese Academy of Sciences, 510640 Guangzhou, China. [5]Department of Earth Science & Environmental Change, University of Illinois at Urbana-Champaign, Champaign, IL 61820, USA. ✉e-mail: hujs@sustech.edu.cn; ljliu@mail.iggcas.ac.cn

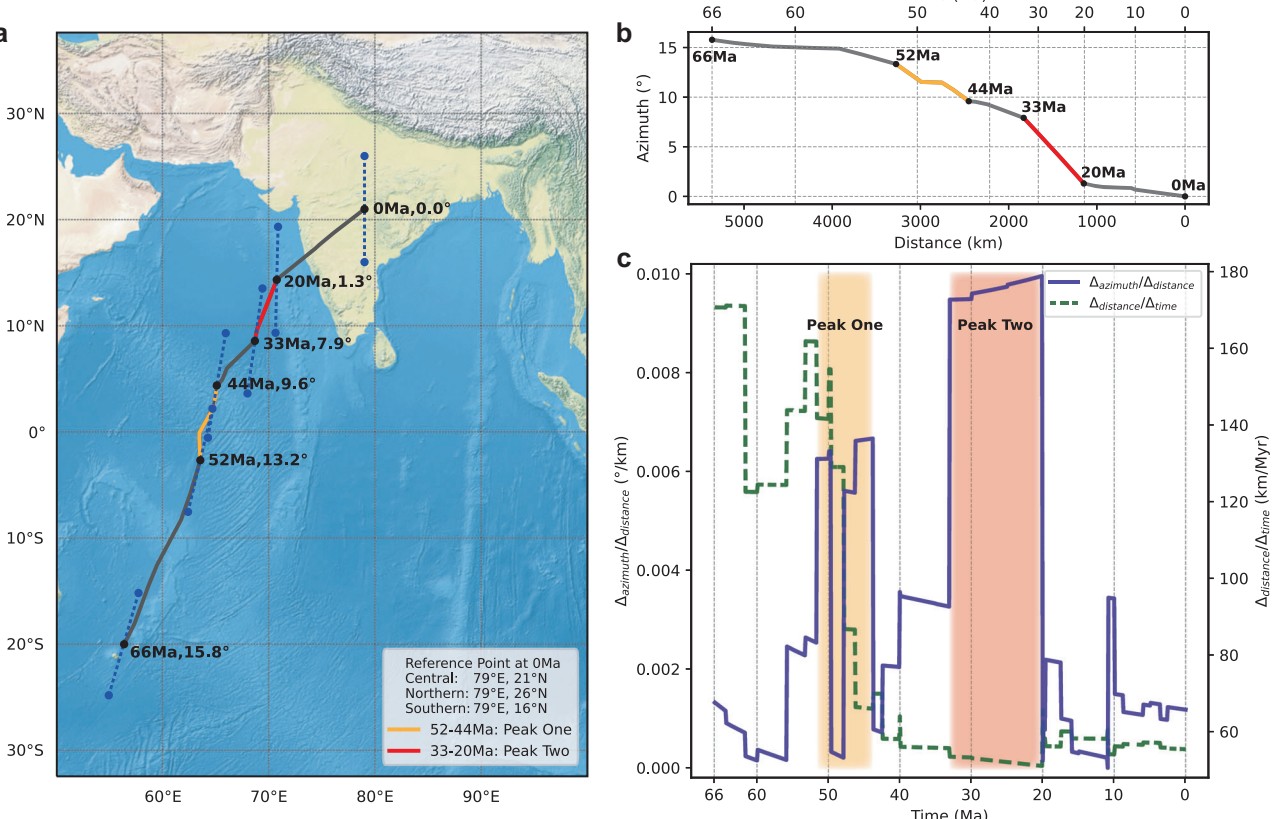

**Fig. 1 | Cenozoic motion of the Indian plate. a** Locations of the central, northern, and southern reference points at 66 Ma, 52 Ma, 44 Ma, 33 Ma, 20 Ma, and 0 Ma, respectively. The dotted blue lines connect the northern and southern points, and their azimuths at specific ages are listed. The black line marks the trajectory of the central point. **b** Variation of the azimuth since 66 Ma. **c** Velocity and rate of rotation since 66 Ma, with two rotation rate peaks highlighted by shading. The motions are calculated based on the plate reconstruction of ref. 15.

collision between India and Eurasia occurred around $55 \pm 5$ Ma in western-central Greater India, and then propagated eastward, with the complete collision occurring around $40 \pm 5$ Ma. We demonstrate that this collision process is consistent with a series of geological evidence in the tectonic domain.

## Results and discussion

### Cenozoic rotational motion of the Indian plate

To illustrate the Cenozoic motion of the Indian plate, we choose three reference points within the Indian sub-continent, which are reconstructed backward in time following a recent global plate reconstruction[15]. The trajectory and velocity of the central point are shown in Fig. 1a and c, respectively, which behave similarly to those reconstructed in previous studies[3-6]. In addition, we obtain the azimuths of the great circles connecting the northern and southern points, the variation of which describes the plate's azimuthal rotation (Fig. 1b). We find that the Indian plate has rotated ca. 15.8° counterclockwise since the early Cenozoic, also consistent with previous estimates[14]. The rate of rotation exhibits two prominent peaks during the Cenozoic (Methods and Fig. 1c). The first peak occurred between ca. 52 Ma and ca. 44 Ma, accompanied by rapid plate deceleration, while the second peak occurred between ca. 33 Ma and ca. 20 Ma with an average plate speed close to that of the present. The rotation rate reached ca. 0.006°/km and ca. 0.010°/km during these two peaks, respectively, significantly faster than the mean background rotation rate of ca. 0.003°/km during Cenozoic (Supplementary Fig. 2d).

Besides the plate reconstruction of ref. 15, we further consider three other global reconstructions[16-18] in estimating the Cenozoic

motion of the Indian plate. These reconstructions differ not only in their relative plate circuits, but also in the absolute motions due to different reference frames (Supplementary Table 1). Interestingly, the resulting rotation patterns are generally similar, with slight differences in magnitude (Supplementary Fig. 2a–c). The first peak in the first model[18] (Supplementary Fig. 2a) is not as robust as those in the other models, likely due to the discrepancy in the adopted absolute reference frames based on either the moving hotspots frame (e.g. ref. 15) or a slab-fitting scheme assuming vertically sinking slabs in the lower mantle[21], where the latter was challenged by recent geodynamic modeling[22]. If the slab-fitting reference frame is replaced by the moving hotspots frame (e.g. ref. 17), the first peak in the first model will also appear similar to that derived from ref. 17. The additional rotation rate peak at ca. 60 Ma in some reconstructions (Supplementary Fig. 2a, b) is not considered in this study, since it is not observed in all reconstructions. The average of the rotation rate curves based on all four reconstructions exhibits two clear peaks (Supplementary Fig. 2d), where both the timing and magnitudes of these peaks remain almost the same as those in ref. 15. Consequently, we deem the two-peak pattern to be a reliable feature of the Cenozoic rotation history of the Indian plate.

### Mechanisms for the variable rotation rates

To investigate the mechanisms that control the Cenozoic rotation of the Indian plate, we construct three-dimensional numerical models using the finite element code CitcomS following our recent effort[23]. A free-subduction system is designed where the Indian plate subducts beneath the Eurasian plate with an initial oceanic slab dipping into the mantle. To test the collision process, three different geometries of

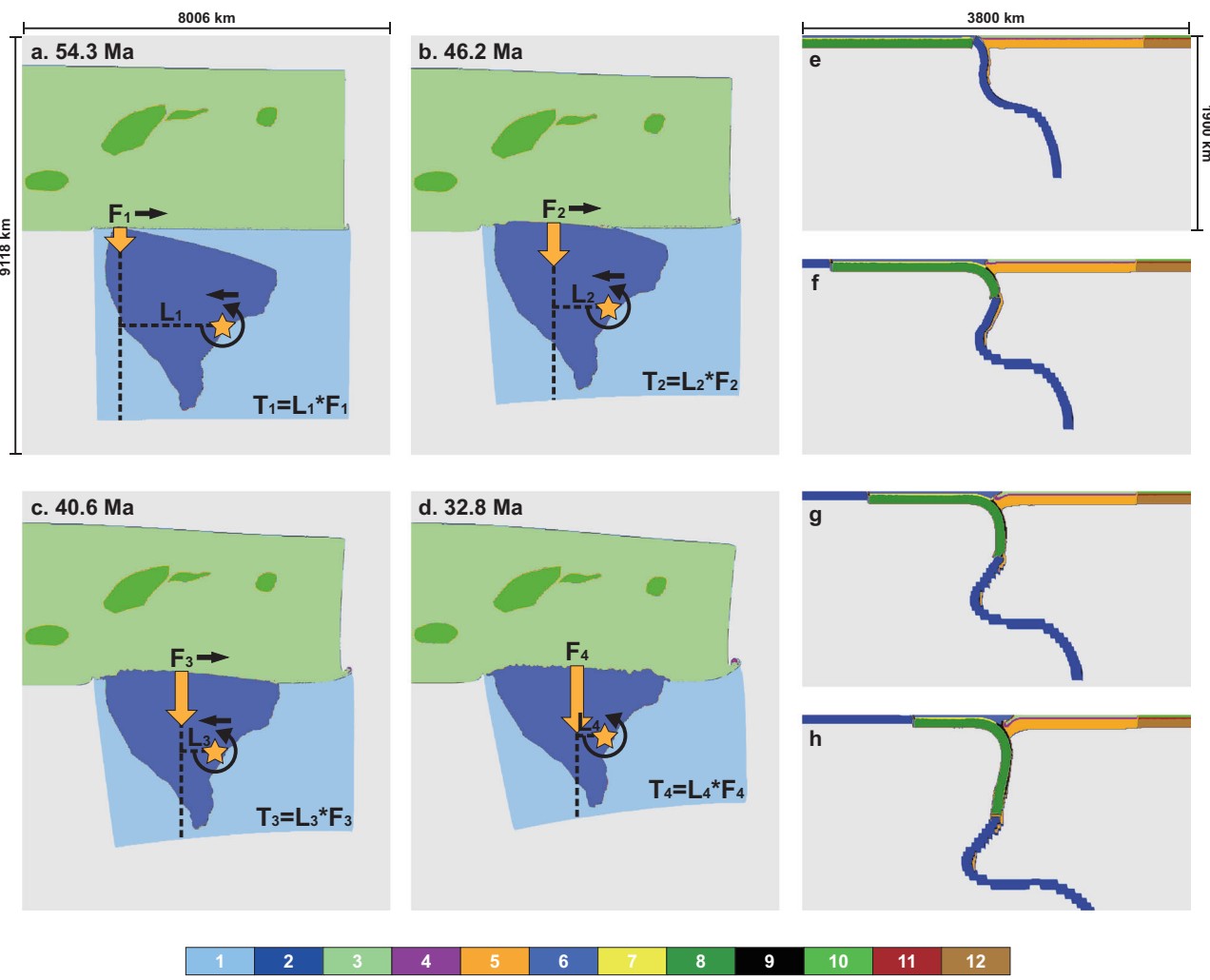

**Fig. 2 | Key snapshots of the evolution of the west-bulging model.** Map view at (**a**) the initial collision, (**b**) the first rotation rate peak, (**c**) the complete collision, and (**d**) the second rotation rate peak. The orange stars represent the center of mass of the Indian plate. The orange arrows represent the resultant resistive force, with their lengths indicating force magnitudes. The horizontal dotted lines are lever arms of the torque. **e**–**h** shows the corresponding zoomed vertical cross-sections cutting through the bulge. Different colors correspond to different compositions:

1-oceanic crust; 2-oceanic mantle lithosphere; 3-Eurasian non-cratonic continental upper crust; 4-Eurasian non-cratonic continental lower crust; 5-Eurasian non-cratonic continental mantle lithosphere; 6-Indian continental upper crust; 7-Indian continental lower crust; 8-Indian continental mantle lithosphere; 9-continental and oceanic eclogitic lower crust; 10-Eurasian cratonic upper crust; 11-Eurasian cratonic lower crust; 12- Eurasian cratonic mantle lithosphere.

Greater India are considered, with a western bulge, middle bulge, and eastern bulge along its northern margin, respectively (Supplementary Fig. 3). Recent regional plate reconstructions[24,25] suggest that the abutting oceanic lithosphere on the eastern side of the Indian continent is much wider than that on the west, and that both oceanic portions are separated from neighboring plates via weak spreading ridges or transform faults. This implies a continuous subduction history along the eastern side of the Indian plate during the Cenozoic, consistent with the recorded continuous arc magmatism in Sumatra[26]. A straight trench geometry (rotated to be E-W oriented along with the Indian plate, see Methods) is adopted at the model's initial condition, as suggested from the along-strike paleolatitudes of the Lhasa terrane[27,28], the restoration of intra-Asian shortening and extrusion[19], and the existence of the linear high-velocity seismic anomaly in the lower mantle[29]. The overriding plate is fixed relative to the western boundary but free to deform on the eastern side, mimicking the rigid Eurasian continental blocks on the west and a relatively weak boundary on the east due to the Pacific and the Neo-Tethys subduction[30,31]. More details of the model setup are discussed in the Methods (Supplementary Figs. 3 and 4).

We simulate the collision process and monitor the rotation of the Indian plate. In the west-bulging model, Greater India initially collides with Eurasia on the western side and then sutures diachronously with Eurasia from west to east (Figs. 2a–d and 3b). The buoyant upper crust of the downgoing Greater India is scraped off along the suture, and its dense eclogitic lower crust and mantle lithosphere subduct continuously (Fig. 2e–h). In this model, the Indian plate experienced two periods of faster rotation, with the first reaching a peak rate of 0.0067°/km and the second of 0.0095°/km (Fig. 3a). Remarkably, both the timing and magnitudes of the two rotation rate peaks are consistent with the observation (Fig. 1c).

The two modeled rotation rate peaks postdate the respective initial collision and complete collision by about 8 Myrs (Fig. 2a–d), reflecting their formation mechanism. The reasoning can be explained by a simple rotational torque analysis. We take the Indian plate as the frame of reference and the geometric center of the Indian plate as its center of mass, which is located next to the southeastern margin of the Indian continent. Assuming the along-strike variation of slab pull (per unit length along strike) is negligible, the collisional resistance will be the main force that influences the rotation of the Indian plate. During

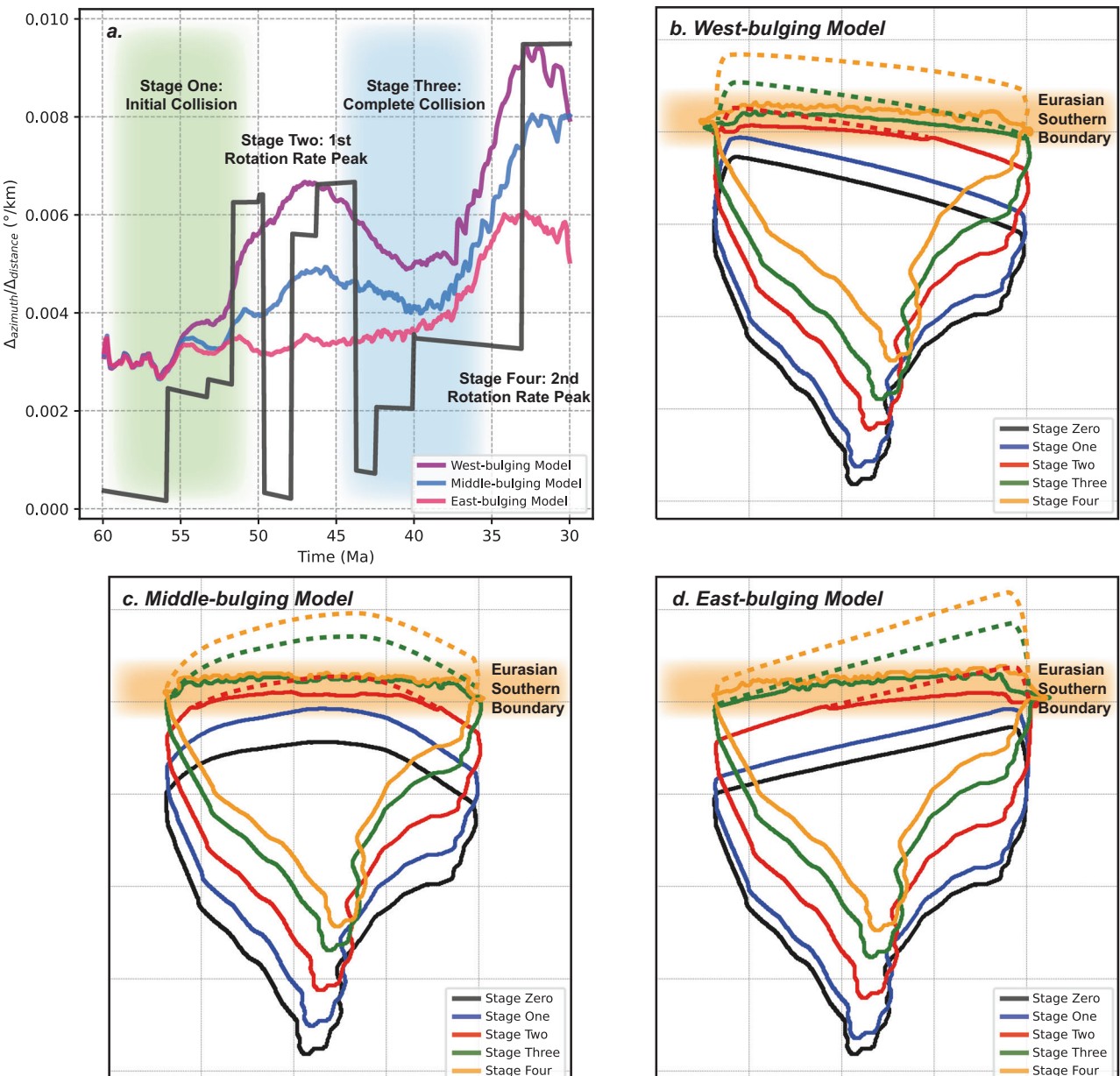

**Fig. 3 | Rotation rates and evolution of the three models. a** Comparison of the rotation rate between the west-, middle-, and east-bulging numerical models and the reconstruction[15]. The possible time intervals of the initial collision and the complete collision are highlighted by shading. **b–d** Evolution of the Indian continent in the west-, middle- and east-bulging models in map views, respectively. Stage Zero corresponds to the initial model setup. The dashed lines mark the outlines of the subducted portion of Greater India.

the initial collision, when the crusts of both continents first come into contact, the resistive force (F), assumed to be proportional to the length of the collisional front, and the rotational torque (T), a product of F and the length of the lever arm (L) relative to the center of mass, are minimal (Fig. 2a). This results in no noticeable extra rotation. Subsequently, the rapidly increasing length of the collisional front boosts F, whose effect dominates that of the decreasing L in determining T (Fig. 2b). The corresponding growing torque accelerates rotation, forming the first rotation rate peak. As it proceeds to a complete collision along the entire Greater Indian front, the decreasing lever arm L outweighs the increasing resistive force F (Fig. 2c), temporally reducing the rotation rate. This could also be verified through a simplified quantitative torque calculation ("Methods" section and Supplementary Fig. 5), where the torque T increases and reaches a peak first, followed by a decrease in T before the collision is

complete. However, after the complete collision, F increases significantly due to the growing amount of accreted crust (thus gravitational potential energy or GPE), upper plate deformation, and inter-plate coupling (Fig. 2f–h and Supplementary Fig. 4b–d), causing T to enhance again (Fig. 2d), which generates the second peak of rotation rate.

For the middle-bulging model, the India-Eurasia collision occurs initially in the central part and then propagates both to the west and the east (Fig. 3c and Supplementary Fig. 6a–d). This model also exhibits two rotation rate peaks, with the first peak at a rate of 0.0049°/km and the second peak at 0.0080°/km (Fig. 3a). The two peaks postdate the initial and complete collision by about 6 Myrs, respectively. The amplitudes of the two peaks are slightly lower than those of the corresponding peaks in the west-bulging model. For the first peak, this is mostly because the length of the lever arm is shorter than that of the

west-bulging model (Supplementary Fig. 6b). This explanation is also supported by the simplified torque calculation, in which the maximum torque (corresponding to the first peak) gets smaller when the bulge moves towards the east (Supplementary Fig. 5b). For the second peak, the difference in amplitude is likely caused by the E-W asymmetry in the distribution of resistive force and possibly slab pull force as well (Supplementary Fig. 6d). For the west-bulging model, the longer collision duration further west means a greater GPE and amount of upper plate deformation, thus a larger resistance west of the center of mass, relative to that in the middle-bulging case. In addition, it is expected that the slab on the east side of the west-bulging model is longer than that in the middle-bulging model following a larger accumulated rotation in the former, and the resulting greater slab pull further enhances the difference in their torques. However, given the uncertainties in both the reconstructions and the simulations (Fig. 3a and Supplementary Fig. 2), we suggest that the middle-bulging scenario is not necessarily inferior to the west-bulging one in terms of fitting the rotation records.

For a better comparison of different scenarios, we also test an east-bulging scenario where Greater India collides with Eurasia on the east first and then sutures diachronously towards the west (Fig. 3d and Supplementary Fig. 7a–d). This model exhibits only one rotation rate peak at a rate of 0.0061°/km, which is significantly lower than that observed (Fig. 3a). The first peak disappears in this model because the position of the initial collision is on the eastern side of the center of mass (Supplementary Fig. 7a), where the resulting clockwise torque does not allow an extra counterclockwise rotation. This clockwise torque is minor in magnitude and does not last long as the suturing propagates westward. The direction of the resultant resistive force is almost in line with the center of mass as the collision develops to Stage Two (Supplementary Fig. 7b), thus no apparent resistive torque exists to create any prominent extra rotation or the observed first rotation rate peak. The simplified quantitative calculation shows a similar tendency of torque variation when the bulge is located on the eastern side (Supplementary Fig. 5d). This model does generate the second observed peak, but with the lowest rate among all three cases (Fig. 3a), due to the similar reasons proposed in the middle-bulging case, including the E-W asymmetry in the distribution of the resistive and slab pull forces. Apparently, this model fits the reconstruction the worst. Therefore, an east-bulging Greater India is the least plausible scenario.

## Implications for plate rotation and Paleogene India-Eurasia collision

Our proposed mechanism for the Cenozoic Indian abnormal rotation due to prominent torques could be further verified from the Mesozoic plate kinematics. For example, the Cretaceous Indian motion during its breakup from Gondwana[32] also exhibits three rotation rate peaks, which occurred at ca. 140-121 Ma, ca. 100-83 Ma, and ca. 73-68 Ma, respectively (Supplementary Fig. 8a). The earliest peak corresponded well with the ca. 136-126 Ma separation between India and Australia-Antarctica[32], likely triggered by the early eruption of the Kerguelen plume at the northeastern Greater India that formed the Comei large igneous province[33] (LIP) at ca. 132 Ma. The second peak was consistent with the ca. 94-84 Ma breakup between India and Madagascar[32] and the emplacement of the ca. 90 Ma Morondava LIP at Madagascar and southwestern India[34]. Similarly, the last peak was coeval with the ca. 71-62 Ma separation of Laxmi-Seychelles from India along its southwestern margin[35] and the ca. 65 Ma eruption of the Deccan LIP[34]. The ca. 10 Myrs gap between the beginning of the rotation rate peaks and the LIPs eruption could be ascribed to the early effect of plume push before the plume head breaks through the lithosphere[5]. The localized plume push applied at the northeastern or southwestern margin of the Indian continent (Supplementary Fig. 8b) exerted large enough torques to have accelerated the counterclockwise rotation of the plate[18].

Mechanically, these plume-generated torques are similar to those due to the diachronous continental collision in driving the Indian rotation. The remarkable correlation between each rotation rate peak and an identifiable geodynamic torque since the early Cretaceous (Fig. 1 and Supplementary Fig. 8) strongly supports the diagnostic implication of the observed Cenozoic Indian rotation on the history of India-Eurasia collision.

Given the good match of our modeled rotation history with that observed (Fig. 3a) and the fact that both the initial and complete collision stages predate the two rotation rate peaks by 5–10 Myrs (Figs. 2 and 3a and Supplementary Figs. 6 and 7), we propose that the India-Eurasia collision commenced at ca. 55 ± 5 Ma along the western-central Tibetan margin and that the complete collision occurred at 40 ± 5 Ma when the collision front expanded to the eastern margin (Figs. 3a and 4). These age ranges are consistent with various recent geological findings along the suture. In southern Tibet, sediment provenance changes have been reported in both deep-water strata of the northern Tethys Himalaya[36,37] (Saga in Fig. 4a) and shallow-water strata of the southern Tethys Himalaya[38] (Tingri in Fig. 4a), giving a collision age of ca. 59 Ma and ca. 51 Ma, respectively. In the western Himalaya, the same approach applied in shallow-water strata[39] constrained the age of collision to be 54 Ma. Ultrahigh-pressure eclogite has also been reported in the western Himalaya[40] with a peak-metamorphic age of ca. 47 Ma, suggesting a collision age at ca. 51 Ma. Further west in Pakistan, it has been proposed that the initial collision occurred at ca. 56-55 Ma based on provenance shift and tectonic responses[41]. Ultrahigh-pressure eclogite in this area with the age of peak-metamorphism at ca. 46 Ma has been reported, where the inferred collision age is similar to that in the western Himalaya[42]. In the eastern Himalaya, however, the collision age has not been well constrained due to the lack of stratigraphic record. High-pressure granulite from this region underwent prograde metamorphism from ca. 40 Ma, implying a similar collision age[43]. Further south in Myanmar, the collision of India-Burma with Eurasia commenced between the late Eocene and the early Oligocene based on ages of the unconformity[44] and provenance changes[45]. In summary, although the geologically implied onset of the India-Eurasia collision varies along the suture, the ages generally fall between ca. 55 ± 5 Ma and ca. 40 ± 5 Ma, closely aligning with our model predictions (Figs. 3a and 4).

Various estimated ages of collision along the strike leave room for discussing the necessity and pattern of diachronous collision. Some studies believe that there is no diachroneity based on the comparison of stratigraphic records distributed in the central-western southern Himalaya[2,46,47], but the collision in the eastern Himalaya and Myanmar is not constrained. Others support the existence of diachroneity, but with different preferences on the detailed collision process. Among them, some argue for an initial collision in the west, based on the earlier cessation of marine sedimentation and the earlier peak metamorphism in the west than those further east[7,8]. However, the inferred collision ages at different regions may not be intrinsically comparable due to the distinct characteristics of utilized rock samples[43]. Some recent studies based on multiple geological indicators along the entire suture proposed that the initial collision occurred in the center[9,10]. This logic, however, suffers from the fact that different indicators from the same region also produce variable ages of collision[2]. Besides, a recent study favored the earliest initial collision in the east based on the deformation feature in southeastern Tibet[12]. However, the deformation along the western-central Tibet was neglected in their discussion, casting doubts on the true along-strike variation implied from the analysis.

Utilizing three-dimensional fully dynamic numerical modeling, we find that the rotation patterns of the west-bulging and middle-bulging Greater India scenarios fit the reconstruction well (Fig. 3a). Based on the observed plate kinematics, our geodynamic analysis, and the existing geological evidence, we propose that the India-Eurasia

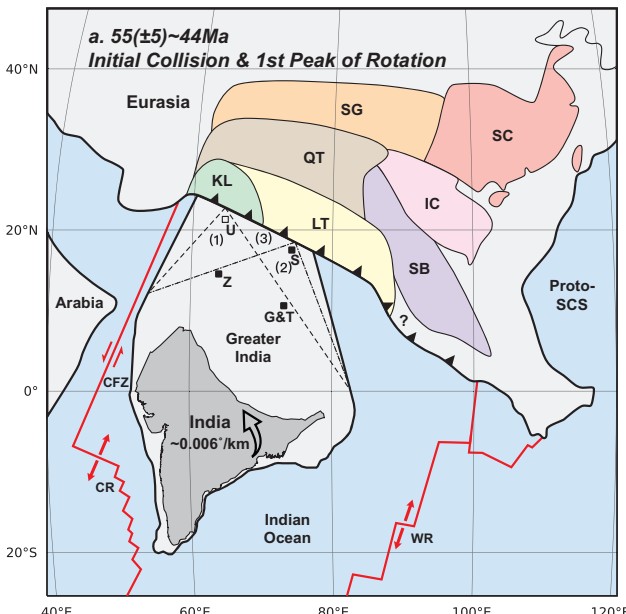

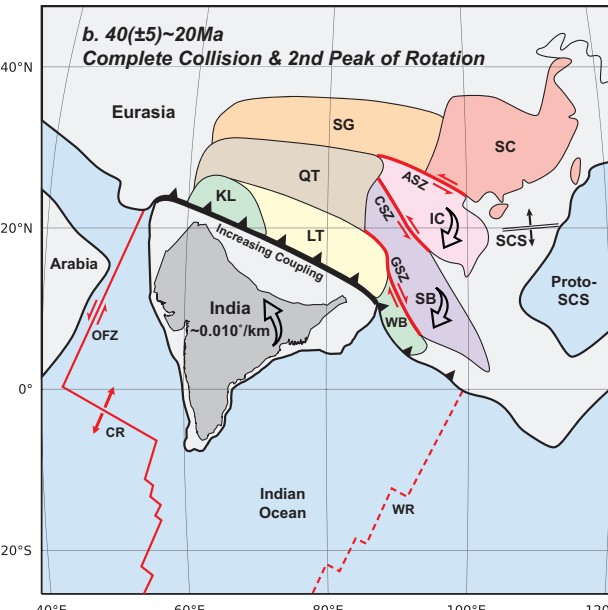

**Fig. 4 | Schematics showing the India-Eurasia collision process. a** The initial collision might have occurred along the (1) western front, (2) central front, or (3) western-central front (coevally) at ca. 55 ± 5 Ma. The first peak of the Indian rotation rate that occurred at ca. 52-44 Ma corresponds to the diachronous collision from west to east. SG Songpan-Ganzi terrane, SC South China block, QT Qiangtang terrane, IC Indochina block, SB Sibumasu block, KL Kohistan-Ladakh arc, LT Lhasa terrane, SCS South China Sea, CFZ Chain Fracture Zone, CR Carlsberg Ridge, WR Wharton Ridge, S Saga area, Z Zanskar area, G&T Gamba and Tingri areas, U

Possible distal area with undiscovered deep-water strata in the western Himalaya[2]. **b** The complete collision along the eastern front might have occurred at ca. 40 ± 5 Ma. The second peak of the Indian rotation rate that occurred at ca. 33-20 Ma was coeval with the significant extrusion of southeastern Tibet and seafloor spreading of the South China Sea, both reflecting strong coupling between the two plates. WB West Burma block, ASZ Ailaoshan shear zone, CSZ Chongshan shear zone, GSZ Gaoligong shear zone, OFZ Owen Fracture Zone. The positions of terrains are referenced from refs. 2,15,19,44,58,70.

collision could have either (1) initiated in the west and then propagated to the east, or (2) initiated in the center, then expanded to the western and eastern side (Fig. 4a). Synchronous initial collision in the central-western Indian front could also be a plausible option (label (3) in Fig. 4a), in which the central and western inner-margin areas (Gamba-Tingri and Zanskar in Fig. 4a) are considered to collide almost coevally, thus the central and western distal-margin areas (Saga and the western area with undiscovered deep-water strata in Fig. 4a) are believed to collide with Eurasia simultaneously[2]. According to our calculation (Supplementary Fig. 5), the first rotation rate peak will appear after suturing around or more than half of the whole Indian front. The distance between Gamba-Tingri and Zanskar (thus between Saga and the undiscovered area) is around one-third of the whole front (Fig. 4a), making it possible to allow two rotation rate peaks. The time interval between the initial collision and the first peak in this scenario will be shorter than in the bulging ones, but still lies in the range of uncertainty and thus implies a similar age of initial collision. These scenarios with western-to-middle-bulging Greater India are compatible with the inferred collision ages discussed above and the widely accepted collisional process suggested by earlier studies[2,7–11,46,47]. Several recent paleomagnetic studies sampling along the Tethys Himalaya delineate the shape of Greater India and suggest that its western-central margin extended farther north than its eastern margin[20,28], further supporting our geodynamically inferred diachronous collision.

While the first rotation rate peak correlates well with syn-collisional processes, the second rotation rate peak during ca. 33-20 Ma (Fig. 1c) might correspond to a special post-collisional stage. All three numerical models exhibit rotation rate peaks during this period (Fig. 3a), which is mainly ascribed to the increasing coupling between the two plates after the final suturing, consistent with the gradual uplift of Himalaya and growing GPE during Oligocene[11]. We notice that this strong plate coupling corresponds well with enhanced tectonic deformation in southeastern Tibet[48,49]. The two major continental

blocks in this region, Indochina and Sibumasu, underwent extrusion of hundreds of kilometers following a clockwise rotation of dozens of degrees[48] (Fig. 4b). The ages of slip along the shear zones between blocks have been constrained to range from early Oligocene to early Miocene[48,49], coeval with the post-collisional rotation rate peak of this study. In addition, the seafloor spreading of the South China Sea[50] initiated at ca. 33 Ma and terminated at ca. 16 Ma, which might also be related to the contemporaneous extrusion of these blocks[30,31] (Fig. 4b). We suggest that the second peak of the Indian plate rotation rate and the intense deformation within southeastern Tibet during ca. 33-20 Ma both reflected strong plate coupling after the complete India-Eurasia collision, which profoundly influenced the evolution of the Tibetan plateau.

There is a possibility that the India-Eurasia collision was more complex than a single-collision scenario[51]. Some of the proposed collision hypotheses can be compared against the rotation history of the Indian plate. Kohistan-Ladakh and West Burma are regarded in several reconstructions[52,53] as oceanic arcs, which could have formed in a double-subduction system situated within the Neo-Tethys Ocean during the Cretaceous[24]. This system is believed to exert a larger slab pull on the adjacent plate than a normal single-subduction system[54,55], such as that located at Sumatra in the east. If true, there would have been a torque resulting in an additional clockwise rotation of the Indian plate until the arc-continent collision eliminated the asymmetry in slab pull. In this case, a continuously increasing rotation rate (assuming negative for clockwise rotation and positive for counter-clockwise rotation) is expected from the weakening of this clockwise torque, followed by the strengthening in counterclockwise rotation induced by continent-continent collision. The Cenozoic double-peak rotation rate pattern (Fig. 1c) seems inconsistent with the scenario where arc-continent collision happened after ca. 60 Ma (e.g. refs. 56). However, arc-continent collision prior to ca. 60 Ma (e.g. refs. 57) is a plausible scenario, although its rotation rate effect might have been

obscured by the influence of plume push in the late Cretaceous (Supplementary Fig. 8). Besides the arc-continent collision, several other configurations of Greater India have been proposed, which generally assumed that this region had undergone significant extension before the collision[19,58]. We suggest this would not change the major modeling results, since the stretched continental lithosphere or the new oceanic lithosphere is still more buoyant than the surrounding old oceanic lithosphere[58], thus more resistive upon collision. Therefore, we speculate that Greater India with a normal or stretched lithosphere are both compatible with the observed rotation features.

To conclude, the Cenozoic rotation motion of the Indian plate could be a novel constraint on the otherwise elusive history of the Paleogene India-Eurasia collision. Among the two prominent counterclockwise rotation rate peaks, the one during ca. 52-44 Ma could reflect the diachronous collision that initiated in the western-central front at 55 ± 5 Ma and then sutured towards the east until 40 ± 5 Ma. The second rotation rate peak during ca. 33-20 Ma should have recorded the formation of strong plate coupling after the final suturing, which drove the deformation of southeastern Tibet. This model of Paleogene India-Eurasia collision is consistent with various geological observations across the tectonic domain. These results suggest that plate rotation history has strong diagnostic implications on abnormal plate kinematic events and the underlying forces and torques that drive these events.

## Methods

### Rotation rates calculations

We calculate the Indian plate rotation rates by measuring the azimuthal changes of the great circle that connects the northern and southern points normalized by the distances that the plate traveled (blue lines in Fig. 1c). This definition of rotation rate is warranted for multiple reasons. First, it can mitigate the effect of the fast time-varying velocity of the India plate (dotted green lines in Fig. 1c), which has an inherent azimuth variation component, except in special circumstances, such as when the Euler pole is located at the North/South pole, or at the equator 90° of longitude away from the central reference point[59]. Second, the Cenozoic Indian plate rotation concerns mostly the dynamic effects of the east-west asymmetry of the downgoing plate at different stages of subduction, which is irrelevant to how fast the plate traverses (Torque calculations in Methods).

### Numerical model

We use the 3D spherical finite element code CitcomS[60] to simulate plate subduction and continental collision. The code solves for thermal-chemical convection governed by the conservation of mass, momentum, and energy, with the assumption that the mantle is incompressible and satisfies the Boussinesq approximation.

The model covers a domain of 72° × 82° × 2890 km in longitude×latitude×depth, which is discretized into 768 × 768 × 128 elements. The size of elements varies spatially, with the finest resolution of 10 × 8 × 8 km occurring in the vicinity of the subduction and collision zones. The model's inner domain, with a size of 50° × 72° × 2890 km, covers parts of the eastern Asian continent, the Indian continent, the Neo-Tethys Ocean, and the Indian Ocean before the continental collision. All features are extracted from a recent plate reconstruction[15] with necessary simplifications for the initial condition (Supplementary Fig. 1), and are rotated counter-clockwise to ensure the model's N-S orientation is parallel to the direction of plate convergence whose actual azimuth falls between ca. 20°−40° in plate reconstructions[15–18]. The azimuth of the convergence boundary was ca. 290°− 310° estimated from the geological proxies[19,27–29], largely perpendicular to the direction of convergence. Thus the trench is correspondingly rotated to E-W direction in the models. A weak region beyond the inner domain is applied to provide enough room for Indian rotation and to avoid artificial return flow from side walls. All

boundaries of the model are free to slip which allows for natural subduction and realistic rotation (Supplementary Fig. 3). The basic model setup is similar to that of ref. 23.

Three-dimensional depth-, temperature-, strain-rate-, and composition-dependent viscosity are considered in the model. A four-layer background viscosity profile is assumed here, including the lithosphere, asthenosphere, transition zone, and lower mantle. Their absolute viscosity values in the models are on the order of $10^{22}$–$10^{23}$, $10^{19}$–$10^{20}$, $10^{19}$–$10^{20}$, and $10^{22}$ Pa·s, respectively. Thin weak layers are applied at oceanic subduction interface to mimic dehydration weakening[23], whose absolute viscosity is set to $8 \times 10^{18}$ Pa·s (Supplementary Fig. 4). The generalized viscosity law used here is

$$\eta(T, r) = A(r)\eta_c \dot{\epsilon}_{II}^{\frac{1-n}{n}} \exp\left( \frac{E_a(r)}{nR\left(T + T_{off}(r)\right)} - \frac{E_a(r)}{nR\left(T_m + T_{off}(r)\right)} \right) \quad (1)$$

where $\eta$ is viscosity, $T$ is temperature, $r$ is radius, $A$ is the pre-factor of viscosity, $\eta_c$ is the pre-factor of composition-dependence, $\dot{\epsilon}_{II}$ is the second invariant of the strain rate tensor, $n$ is the creep exponent, which equals 1.0 for diffusion creep and 3.5 for dislocation creep, $E_a$ is the activation energy, $T_{off}$ is the temperature offset, $T_m$ is the mantle temperature and $R$ is the gas constant. The viscosity of continental lithosphere is controlled by composition tracers considering the complexity of continental geotherm and composition, and its temperature effect is not considered for simplicity. All the depth-dependent rheology parameters are detailed in Supplementary Table 2. We use a relatively low activation energy for diffusion creep compared to values suggested by laboratory experiments[61,62], to account for various slab weakening mechanisms such as slab faulting, hydration, and grain-size damage[63]. The final effective viscosity is controlled by the combination of diffusion creep, dislocation creep, and pseudo-plasticity

$$\eta_{eff} = \min\left( \frac{\sigma_y}{\dot{\epsilon}_{II}}, \frac{\eta_{dif}\eta_{dis}}{\eta_{dif} + \eta_{dis}} \right) \quad (2)$$

where the diffusion creep, $\eta_{dif}$, is applied in the whole domain, while the dislocation creep, $\eta_{dis}$, is only considered in the domain shallower than 660 km. $\sigma_y$ is the yield stress following the Drucker-Prager yield criterion[64,65]

$$\sigma_y = \mu P + C \quad (3)$$

$$\mu = \mu_0 \left[1 - \min\left(1, \frac{\varepsilon_p}{\varepsilon_f}\right)\right] \quad (4)$$

$$C = C_f + (C_0 - C_f)\left[1 - \min\left(1, \frac{\varepsilon_p}{\varepsilon_f}\right)\right] \quad (5)$$

where $\mu$ is the coefficient of friction, $\mu_0$ is the initial coefficient of friction, $C$ is the cohesion, $C_0$ is the initial cohesion, $C_f$ is the minimum cohesion, $\varepsilon_f$ is the reference plastic strain, $\varepsilon_p$ is the accumulated plastic strain after the yield stress is reached. The material gets weaker as plastic strain accumulates. The lateral composition-dependent rheology parameters are detailed in Supplementary Table 3.

The composition field is defined using Lagrangian tracers (Fig. 2, ref. 66). The continental lithosphere has a three-layer structure which consists of the upper crust, the lower crust, and the lithospheric mantle. The Eurasian continent is further divided into the weaker non-cratonic and stronger cratonic regions, with the latter including the Karakum, Tarim, Qaidam, and Sichuan basins. The oceanic lithosphere has a two-layer structure that consists of the crust and the lithospheric mantle. For convenience, the initial temperature fields of the

continental and oceanic lithosphere are both defined using the half-space cooling model, assuming a mantle potential temperature of 1300 °C. The initial age is 50 Myrs for the Indian oceanic plate, 100 Myrs for the Eurasian non-cratonic regions, 150 Myrs for the Indian continent, and 200 Myrs for the Eurasian cratonic regions. Temperature-, composition-, and phase-change-dependent density anomaly is considered in the models. The density anomaly used here is defined as:

$$\delta\rho = -\alpha_0\rho_0\Delta T + \delta\rho_{ph}\Gamma + \delta\rho_{ch} \qquad (6)$$

where $\alpha_0$ is thermal expansivity, which equals $3.0\times10^{-5}$ /K, $\rho_0$ is reference density, which equals 3340 kg/m³, $\Delta T$ is temperature difference, $\delta\rho_{ph}$ is the density jump across the phase change, $\Gamma$ is the phase function[67], $\delta\rho_{ch}$ is the chemical density difference between the compositions. The detailed thickness and chemical density setups for all compositional layers are listed in Supplementary Table 4.

## Torque calculations

We perform a straightforward calculation on the variation of the collision-related resistive torque (Supplementary Fig. 5). We take the Indian plate as the frame of reference. The width of the Indian continent is s, which equals 3500 km in the calculation. The center of mass of the Indian plate initially lies s/3 away from the eastern boundary (represented by parameter b in Supplementary Fig. 5a). Considering the counterclockwise rotation of the Indian plate (e.g. Figure 2) and asymmetric trench geometry due to the continental indentation on the west[30,31] and trench retreat on the east[68], the unsubducted portion of the plate on the west is longer than on the east. The center of mass moves westward linearly as the collision goes on, with the maximum offset reaching 500 km. The southward movement of the mass center of the remaining Indian plate is not considered in the calculations, since this does not affect the length of the lever arm and the magnitude of the resistive torque. The three parameters above are estimated according to the recent reconstruction[15]. The location of the bulge varies from 0 to s away from the western boundary (represented by parameter a in Supplementary Fig. 5a). The resistive torques are calculated via the contact width of the continents, thus the effects of the plate's time-varying transverse motion and the latitude-varying locations of the bulge are eliminated. We take the resistive force along the collisional front[69] as $8.0\times10^9$ N/km, and assume the force is proportional to the contact width.

When the bulge lies at the west side of the collision front (a <s/2), the torque increases quickly after the initial collision (contact width = 0 in Supplementary Fig. 5b) and reaches a maximum peak, then decreases gradually (blue, orange, green and dark red curves in Supplementary Fig. 5b). This scenario is similar to the first rotation peak in the west-bulging model (Fig. 3a, b). In the middle-bulging model (a = s/2, purple curve in Supplementary Fig. 5b), due to westward movement of the mass center and associated shortening of the lever arm, its behavior is similar to the west-bulging model but with a lower first peak (Fig. 3a, c). When the bulge lies at the east side of the collision front (a > s/2), the center of mass in the model moves eastward first and then moves westward. Thus the total displacement of the mass center is very minor (Supplementary Fig. 5c). The corresponding calculation is shown in Supplementary Fig. 5d. The torque decreases slightly and reaches a minimum, then it increases quickly (pink, gray, and light green curves in Supplementary Fig. 5d). This scenario is similar to the East-bulging model that does not exhibit the first rotation peak (Fig. 3a, d). We emphasize this torque calculation is a first-order simplification of model results. The torque of east-bulging scenario with a fixed mass center during the full collision (e.g. gray curve in Supplementary Fig. 5d) reaches a larger value than that of west-bulging scenario with a moving mass center (e.g. orange curve in Supplementary Fig. 5b), which seems to be contradictory with the model results (Stage Three in Fig. 3a). This could be ascribed to the increasing resistive force on the

eastern portion of the east-bulging model as collision goes on, which would weaken the final counterclockwise rotation.

## Reporting summary

Further information on research design is available in the Nature Portfolio Reporting Summary linked to this article.

## Data availability

The datasets used in calculating rotation motions displayed in Fig. 1 and Supplementary Figs. 2 and 8 are available from the cited papers.

## Code availability

The computational code CitcomS is available at www.geodynamics.org. The python library pyGPlates that provides the functionality to reconstruct plate motions can be accessed at www.gplates.org.

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

## Acknowledgements

J.H. was partially supported by the National Natural Science Foundation of China (NSFC) through awards 92155307 and 42174106. L.C. was partially supported by the Strategic Priority Research Program (A) of Chinese Academy of Sciences, Grant No. XDA20070302. Computations were supported by Center for Computational Science and Engineering at Southern University of Science and Technology.

## Author contributions

J.H. and L.J.L. conceived and oversaw the project. X.W. carried out the numerical experiments. L.C. and L.L. contributed to geological analysis. All authors participated in result interpretation and manuscript preparation.

## Competing interests

The authors declare no competing interests.
