## [Peer Review File · Nature Communications]

REVIEWER COMMENTS

Reviewer #1 (Remarks to the Author):

This is an interesting and innovative paper which links periods of enhanced rotation of the India plate with the India-Asia collision history. Overall, the proposed scenario makes sense, I think, and, to my knowledge, this link has not been proposed before. The modelling is state-of-the-art, and I also like the simple torque computations which help understanding what is going on, and the paper is quite well-written.

I think there are two main points (and a couple of minor ones) though, where the paper could be improved I think.

My first issue is that I think the result, and in particular the first rotation peak, is not very robust. It occurs in the Müller et al. models which are, I suppose, related as they are both with Müller as first author but cannot clearly be seen in the others (not at all in van Hinsbergen, only very vaguely in Torsvik). Because there are two out of four models from Müller, it can also be seen in the average, but I think this doesn't mean anything. What I am wondering is whether, apart from comparing different models, it is also possible to do a formal error analysis? I mean, plate motions and rotations usually come with uncertainty ellipses, so would it be possible to compute whether these rotation peaks are significant, considering uncertainties?

My other issue is that some of the statements and figures in the paper appear actually contradicting each other. Maybe I misunderstand something here, but it requires clarification: In Figure 3a, all three model curves show counterclockwise (positive) rotation all the time. But in the other three panels, there is only counterclockwise rotation seen in b (west-bulging model), whereas c (middle-bulging model) shows more or less no rotation at all, and d (east-bulging model) actually shows clockwise rotation. And the middle-bulging model presumably corresponds to the $a=s/2$ case in Extended Data Figure 5. But in contrast to Fig. 3a, this shows no peak but just a continuous increase. And also the East-bulging model looks different again in Extended Data Figure 5: Here the torque is first negative, then becomes positive, whereas in 3a it is always positive and in 3c always clockwise (i.e. negative). So, I am confused.

Further comments:

line 114: But, North-South convergence, as described in the Methods (l. 499/500) is not identical to E-W boundary, unless the convergence is orthogonal to the boundary. Is that the case?

line 238: Isn't the pre-dating by 5-10 Myrs best seen in Fig. 3a?

line 265: Why is Sibumasu Block abbreviated with "SP"? What does the "P" stand for?

line 482: "as long as the Euler Poles do not overlap with the North Pole" - I am not sure what you mean with this. I mean, also if there is an Euler pole at the equator 90° away from the continent, this corresponds to pure northward motion with no rotation.

For the absolute viscosity values in line 508, do you mean in reality or in your model? I am also wondering whether you need to specify a higher viscosity in the lithosphere; doesn't that come in automatically with the higher temperature, so you don't need to explicitly prescribe a higher viscosity (because otherwise, the same effect is considered twice)?

In your viscosity law in line 512, everything seems to be dimensionless, so how about the second invariant of the strain rate tensor, this should normally be in units of 1/s; how do you get this dimensionless?

line 571: I think the sentence "The width ..." can be improved. I prefer how it is said on line 109/110 "oceanic lithosphere on the eastern side of the Indian continent is much wider than that on the west"

Extended Data Figure 5: Why is there a "1e16" on top of panel b? Why you don't include curves for a=0 and a=1?

Extended Data Table 3: So here you don't distinguish between Upper Crust, Lower Crust and Lithospheric Mantle? An the distinction between Asian Non-Cratonic and Cratonic Lithosphere is for all three (Upper Crust, Lower Crust and Lithospheric Mantle)?

Extended Data Table 4: Does "Neutral" mean it is the same as the underlying mantle, apart from Temperature dependence? And what is actually the temperature dependence of density (i.e. thermal expansivity)? I don't see it anywhere specified.

Minor commenst:

line 33: "has been debated"

line 294: "could have either"

Reviewer #2 (Remarks to the Author):

This is an interesting paper bringing some new constraints to understanding the history and potential driving forces of the India-Eurasia collision. Using the connection between changes in the azimuth of the India-Eurasia collision to constrain geodynamic models is a novel way of looking at this problem. I enjoyed reading the paper, and suggest that it should be published with some relatively minor corrections. The English language needs a bit of improvement throughout the paper. I have made some comments along these lines.

In terms of the interpretation of the results, I have this overall comment: The authors are following the premise that the time-dependent changes in rotation of India relative to Eurasia can only be explained by a successive continent-continent collision which needs to start around 55Ma or earlier, necessitating an at least 2000 km wide thinned continental margin off northwestern India (the details of the inferred geometry are difficult to extract from the paper).

The first question one needs to ask oneself is: Is this geodynamically reasonable or possible? The widest continental margin in existence today is about 400 km in width. There is a simple reason for this. It is not rheologically possible to create a much wider passive margin, as the stretched lithosphere will simply break, producing seafloor spreading, preventing excessive stretching from occurring. Stretching factors larger than four lead to SFS (see classical papers by Dan McKenzie). Continental crust is too weak to be stretched beyond this limit. So how exactly do you propose to generate a 2000 km wide passive continental margin to start with, 5 times wider than any margin in existence? In my view this essentially falls in the land of geodynamic phantasy, so I'd like to see this justified in the paper (personally, I don't think this is possible, but I am looking forward to see the explanation you come up with!). The subduction of this much continental crust also presents other problems, as is discussed, for instance, in van Hinsbergen DJ, Lippert PC, and Huang W. Unfeasible subduction? *Nature Geoscience*. 2017 Dec;10:878-9.

Along the same lines, I am surprised that the recent paper by Pusok and Stegman (The convergence history of India-Eurasia records multiple subduction dynamics processes, *Sci Adv* 2020) is not cited at all. Essentially it provides some ideas for an alternative way to create a two-stage collision history, with the first collision reflecting the collision of an island arc with Eurasia followed later by a continent-continent collision. This scenario was also proposed in a number of other papers. This should at least be discussed as a potential alternative way to explain the kinematic observations.

Some detailed comments:

Line 16:

The Cenozoic India-Eurasia collision has profound impacts on the formation of the modern Tibetan plateau.

should be

The Cenozoic India-Eurasia collision has had profound impacts on the formation of the modern Tibetan plateau.

Line 136, and many subsequent instances:

I do not understand why in " West-bulging model", "Middle-bulging model" etc "west" and "middle" is capitalized. It should be lower-case.

Line 145:

The two modeled rotation peaks postdate the respective initial collision and complete collision by about 8 Myrs (Fig. 2a-d), which bears their formation mechanism.

should be

The two modeled rotation peaks postdate the respective initial collision and complete collision by about 8 Myrs (Fig. 2a-d), reflecting their formation mechanism.

Line 150: neglectable

should be

negligible

Line 154: This results in no noticeable rotation.

The use of the word rotation is ambiguous here, as well as through much of the rest of the paper. The issue is that all plates rotate about Euler poles at all times, ie plates are always in rotation. What you mean here is rotating about a pole located proximal to the Indian continent. You need to think about how to do that better, ie you cannot write that at some point India was not rotation, and then it started to rotate, seeing that it is ALWAYS rotating.

Line 166: Similarly, a "peak in rotation" is not the correct term here. You actually mean a peak in rotation rate. The same issue occurs in other places in the text.

Line 261: Some of the letter labels overlying "Greater India" are barely visible, especially those frequently referred to in the text like the "Saga" and "Tingri" areas.

Response to Reviewers' Comments

In this response, we provide point-by-point responses (in blue) to all the comments (in black). The revised manuscript is marked in red and green displaying the revisions.

Reviewer #1 (Remarks to the Author):

This is an interesting and innovative paper which links periods of enhanced rotation of the India plate with the India-Asia collision history. Overall, the proposed scenario makes sense, I think, and, to my knowledge, this link has not been proposed before. The modelling is state-of-the-art, and I also like the simple torque computations which help understanding what is going on, and the paper is quite well-written.

I think there are two main points (and a couple of minor ones) though, where the paper could be improved I think.

Thank you very much for taking the time reviewing our manuscript. We appreciate your positive comments and valuable suggestions on our paper that have greatly improved our manuscript.

(1) My first issue is that I think the result, and in particular the first rotation peak, is not very robust. It occurs in the Müller et al. models which are, I suppose, related as they are both with Müller as first author but cannot clearly be seen in the others (not at all in van Hinsbergen, only very vaguely in Torsvik). Because there are two out of four models from Müller, it can also be seen in the average, but I think this doesn't mean anything.

This is a good point. We have re-examined the rotation parameters of the four plate reconstructions and tested the effects of different relative plate circuits and absolute plate motions (Extended Data Table 1). As for the relative plate circuits used by van Hinsbergen et al. (2021) and Torsvik et al. (2019), they are actually based on the same reference papers for relative plate motions, with few differences in geologic time scale. The reason why their rotation features behave differently is mainly due to the different absolute motion frames they use. For example, van Hinsbergen et al. (2021) used the slab-fitting absolute frame that reconstructs plate motions assuming slab remnants sink vertically in the lower mantle (van der Meer et al., 2010). This assumption is highly controversial considering the enormous lateral slab displacement caused by horizontal mantle flow (Peng & Liu, 2022). If we combine the plate circuits from van Hinsbergen et al. (2021) with the absolute motion frame used by Torsvik et al. (2019), the "hybrid" reconstruction is similar to those derived from Torsvik et al. (2019) (Fig. R1), thus verifying that the two relative motions in van Hinsbergen et al. (2021) and Torsvik et al. (2019) are indeed similar, both showing a peak during 52-44 Ma, although not as significant as in Müller et al. (2016) and Müller et al. (2019). We prefer the reconstructions using absolute frames based on moving hot spots and various other constraints, such as those proposed by Torsvik et al. (2008), Doubrovine et al. (2012), and Tetley et al. (2019) that are used by the other three plate reconstructions (Müller et al., 2016; 2019; Torsvik et al., 2019). Here, we want to emphasize that except the controversial slab-fitting absolute frame, combining

other absolute frames (e.g. the three mentioned above) and different relative plate motions does generate the peak during 52-44 Ma. We show van Hinsbergen et al. (2021) for completeness. Consequently, the average of the four reconstructions with different relative plate circuits and absolute frames effectively weighs down the contribution from the slab-fitting absolute frame, and thus should better represent the preferred plate motion. We have added some extra descriptions in the main text (lines 94-98).

Fig. R1. The rotation rate and velocity derived from the "hybrid" reconstruction (van Hinsbergen et al. (2021) relative motion + Torsvik et al. (2019) absolute motion) (left) and the Torsvik et al. (2019) reconstruction (right).

What I am wondering is whether, apart from comparing different models, it is also possible to do a formal error analysis? I mean, plate motions and rotations usually come with uncertainty ellipses, so would it be possible to compute whether these rotation peaks are significant, considering uncertainties?

We appreciate your constructive suggestion. We have collected all available covariance matrices of the finite rotations used in the four global reconstructions, which denote uncertainties in reconstructions (Kirkwood et al., 1999). But the covariance matrices of absolute motions are lacking (e.g. Tetley et al. (2019) for Müller et al. (2019), Torsvik et al. (2008) for Müller et al. (2016), van der Meer et al. (2010) for van Hinsbergen et al. (2021)), which prohibits the evaluation of absolute motions of the Indian plate. Therefore, we are unable to convert the segmented rotation uncertainties to a composite uncertainty or to get their confidence region. The calculation for the uncertainties of rotation rates have not been studied before, which is beyond our capabilities and scope of this study. Besides, as suggested by White & Lister (2012), the true uncertainty of reconstruction involves many more factors than are currently taken into account, such as the deformation within the plates and the uncertainties when distinguishing magnetic anomalies. We will relay this effort to future research.

(2) My other issue is that some of the statements and figures in the paper appear actually contradicting each other. Maybe I misunderstand something here, but it requires clarification: In Figure 3a, all three model curves show counterclockwise (positive) rotation all the time. But

in the other three panels, there is only counterclockwise rotation seen in b (west-bulging model), whereas c (middle-bulging model) shows more or less no rotation at all, and d (east-bulging model) actually shows clockwise rotation.

Thanks for pointing this out. We agree that this figure is a bit misleading. We have redrawn Figure 3 to make it clearer. We have added the subducted portion of Greater India at the stages two to four with dashed lines, which better exhibits their trajectory after collision. The northeastern edge of the plate moves faster than the northwestern edge in all three models, which denotes their continuous counterclockwise rotation.

And the middle-bulging model presumably corresponds to the $a=s/2$ case in Extended Data Figure 5. But in contrast to Fig. 3a, this shows no peak but just a continuous increase. And also the East-bulging model looks different again in Extended Data Figure 5: Here the torque is first negative, then becomes positive, whereas in 3a it is always positive and in 3c always clockwise (i.e. negative). So, I am confused.

This is a very good point. The simple torque analysis is really a simplification of the complex collision process simulated by our models. We have updated the torque calculation (Extended Data Fig. 5), which should be more realistic and reasonable than the previous analysis (lines 164-166, 179-182, 204-205, 610-614, 626-643). We notice that the mass center of the Indian plate would move westward after the collision both in the numerical models and in the plate reconstructions. Because of the extra counterclockwise rotation since the collision, the Indian plate remaining on the surface would become wider on the western side than on the eastern side in the numerical models (Fig. 2). Thus the center of mass should move westward along with the collision process. This tendency is also reflected in plate reconstructions (Gibbons et al., 2015; Zahirovic et al., 2016). The eastern boundary of the Indian plate, the Wharton Ridge, intersected with Eurasia from Java and eastern Sumatra to western Sumatra during 55-40 Ma. The displacement of the ridge is estimated to be ca. 1000 km, which corresponded to the ca. 500 km westward movement of the center of mass. Besides, the trench advance in the west caused by continent indentation (Tapponnier et al., 1982; Jiao et al., 2023) and trench retreat in the east due to back-arc extension (Li et al., 2013; Sternai et al., 2014; Capitanio et al., 2015) would also facilitate the asymmetry of the remaining plate [Redacted].

[Redacted]

Then we add this feature into the torque calculation, in which the center of mass moves westward continuously during the collision process (Extended Data Fig. 5, Fig. R3a,b). The results are more consistent with the modeling results, with both the west-bulging scenarios ($a < s/2$) and the middle-bulging scenario ($a = s/2$) exhibiting positive peaks of resistive torques, which correspond to the first peak of the rotation rate (Fig. 3a). Most east-bulging scenarios ($a > 3s/4$) exhibit a negative peak first and then turn to a positive value when it comes to the complete collision. However, we would like to point out that the displacement of the mass center in east-bulging models is relatively small. The original calculation with a fixed mass center should make more sense. The duration of a negative torque and its magnitude should be minor (Fig. R3c,d). Besides, the overriding Eurasian plate is fixed on the west side and free on the east side in the models. The asymmetry-induced clockwise rotation of the Eurasian plate could facilitate the accumulation of the eastern oceanic slab (Extended Data Fig. 7), and the resulting greater slab pull would weaken the initial clockwise rotation. Therefore, the clockwise rotation is not significant in the east-bulging numerical model. And the first-order phenomenon, the absence of positive peaks of torques for the east-bulging scenarios ($a > s/2$), is consistent with the disappearance of the first peak in rotation rates, which precludes the possibility that the first collision occurred on the eastern side.

Fig. R3. Calculation of the resistive torques. a, Diagram of the Indian continent before collision. The center of mass of the Indian plate is set to move westward during the collision process. Other captions are the same as in Extended Data Figure 5. b, Calculated resistive torque versus contact width between continents. c and d are corresponding diagrams and curves considering the center of mass remains fixed.

Further comments:

line 114: But, North-South convergence, as described in the Methods (l. 499/500) is not identical to E-W boundary, unless the convergence is orthogonal to the boundary. Is that the case?

A short answer is Yes. We have considered this when constructing the model. Here, we extracted

the azimuth of the Cenozoic trajectory of the central reference point, which represents the drifting direction of the Indian plate. The average of azimuth spans from 40° to 20° (Fig. R4), which is consistent with that derived by Gibbons et al. (2015). We have also calculated the strike of the trench derived from different proxies. The recent paleomagnetism studies constrains that the azimuth spans from 317° to 290° (Yi et al., 2015; Tong et al., 2022; Bian et al., 2022; Ma et al., 2022). The restoration of the intra-Asian shortening and extrusion shows that the boundary is oriented at around 293° - 291° (van Hinsbergen et al., 2019). The azimuth of the linear high-velocity seismic anomaly in the lower mantle is around 301° - 305° (Replumaz et al., 2014). Overall, the convergence boundary before the collision is oriented at around 290° - 310° . Thus, the convergence is almost orthogonal to the boundary (Fig. R4). For clarification, we have added extra descriptions in the Methods (lines 117-121, 551-554).

Fig. R4. Convergence direction of the Indian plate derived from the four reconstructions (left) and the diagram of the convergence direction and the azimuth of the convergence boundary (right).

line 238: Isn't the pre-dating by 5-10 Myrs best seen in Fig. 3a?

Agree. We have added Fig. 3a as an additional figure reference. See line 244.

line 265: Why is Sibumasu Block abbreviated with "SP"? What does the "P" stand for?

The "SP" here represents Shan Plateau, another widely-used name for this terrane. We have changed the abbreviation here to "SB". See line 271.

line 482: "as long as the Euler Poles do not overlap with the North Pole" - I am not sure what you mean with this. I mean, also if there is an Euler pole at the equator 90° away from the continent, this corresponds to pure northward motion with no rotation.

Thanks for your comment. We have corrected it here. See lines 532-533.

For the absolute viscosity values in line 508, do you mean in reality or in your model?

The values here refer to the absolute viscosity in our models. We have added “in the models” in this sentence. See line 562.

I am also wondering whether you need to specify a higher viscosity in the lithosphere; doesn't that come in automatically with the higher (lower?) temperature, so you don't need to explicitly prescribe a higher viscosity (because otherwise, the same effect is considered twice)?

Thanks for pointing this out. We have added extra explanations in the Methods (lines 570-572). The viscosity of the oceanic mantle lithosphere is generally controlled by temperature in the models, as its temperature and composition (olivine) are relatively well known. But for the continent, its geotherm and compositions are more complex. For simplicity, we mainly consider the effective viscosity for continents, whose value could be easily approximated by the composition tracers of the crust and lithosphere mantle. Therefore, we use a special viscosity setting in the continent, which means that if an element contains composition tracers of continental lithosphere, then its composition-dependent viscosity will take over and the temperature effect will not take effect. In theory, the temperature dependence should also affect viscosity, but this is neglected for the above purpose.

In your viscosity law in line 512, everything seems to be dimensionless, so how about the second invariant of the strain rate tensor, this should normally be in units of 1/s; how do you get this dimensionless?

Yes, the viscosity law here is dimensionless. In the calculation of the second invariant of the strain rate, every parameter including velocity and length is dimensionless, which makes the second invariant of the strain rate dimensionless automatically. The dimensionless factor of time equals R_0^2/κ_0 , in which R_0 is the radius of the Earth, and κ_0 is the reference thermal diffusivity. But if you consider the dimension in the viscosity law, the unit 1/s of the strain rate tensor is canceled out by the unit $s(1_n)tn$ in the prefactor $A(r)$.

line 571: I think the sentence "The width ..." can be improved. I prefer how it is said on line 109/110 "oceanic lithosphere on the eastern side of the Indian continent is much wider than that on the west"

We have changed the expressions here. See lines 648-650.

Extended Data Figure 5: Why is there a "1e16" on top of panel b? Why you don't include curves for a=0 and a=1?

The 1e16 here is the multiplier of the numbers beside the vertical axis. We have redrawn Extended Data Figure 5 and included the curves for a=0 and a=1.

Extended Data Table 3: So here you don't distinguish between Upper Crust, Lower Crust and Lithospheric Mantle? And the distinction between Asian Non-Cratonic and Cratonic Lithosphere is for all three (Upper Crust, Lower Crust and Lithospheric Mantle)?

We have considered the viscosity layering of the continental lithosphere in the original models and corrected the chart accordingly (Extended Data Table 3). The distinction of viscosity between Asian non-cratonic and cratonic lithosphere is for the whole lithosphere.

Extended Data Table 4: Does "Neutral" mean it is the same as the underlying mantle, apart from Temperature dependence? And what is actually the temperature dependence of density (i.e. thermal expansivity)? I don't see it anywhere specified.

We have added the description of density in the Methods (lines 596-603). Temperature-, composition-, and phase-change-dependent density is considered in the numerical modeling. The word "Neutral" here means that the composition-dependent density of the lithospheric mantle is the same as that of the underlying mantle. The thermal density anomaly is defined as $-\alpha_{\rho,0} \Delta T$. The $\alpha_{\rho,0}$ in this expression is the thermal expansivity, which equals $3.0 \times 10^{-5} / \text{K}$ in the modeling.

Minor comment:

line 33: "has been debated"

There are two subjects in the sentence, i.e. "when India collided with Eurasia" and "how this process varied along the strike of the Indian front". Thus we think the word "have" here should be more appropriate.

line 294: "could have either"

Thank you. This has been changed. See lines 300-301.

Reviewer #2 (Remarks to the Author):

This is an interesting paper bringing some new constraints to understanding the history and potential driving forces of the India-Eurasia collision. Using the connection between changes in the azimuth of the India-Eurasia collision to constrain geodynamic models is a novel way of looking at this problem. I enjoyed reading the paper, and suggest that it should be published with some relatively minor corrections. The English language needs a bit of improvement throughout the paper. I have made some comments along these lines.

Thank you very much for your positive comments and valuable suggestions on our paper that have greatly improved our manuscript. We have also improved the English language based on your comments.

In terms of the interpretation of the results, I have this overall comment: The authors are following the premise that the time-dependent changes in rotation of India relative to Eurasia can only be explained by a successive continent-continent collision which needs to start around

55Ma or earlier, necessitating an at least 2000 km wide thinned continental margin off northwestern India (the details of the inferred geometry are difficult to extract from the paper).

We appreciate this brief summary of the reviewer. This is a good point that necessitates a brief discussion of the different modes of India-Eurasia collision, as well as the morphology and position of Greater India during collision, as part of the broad implication of this study. Initially, we mainly focus on the NW-SE asymmetry-induced time-varying resistive torque and rotation rate. Then we further discuss different collision scenarios, where we show that the NE-SW width, crustal thickness, and characteristics (continental or oceanic) of Greater India are subject to heavy debate and that our model is compatible with multiple hypotheses.

(1) The first question one needs to ask oneself is: Is this geodynamically reasonable or possible? The widest continental margin in existence today is about 400 km in width. There is a simple reason for this. It is not rheologically possible to create a much wider passive margin, as the stretched lithosphere will simply break, producing seafloor spreading, preventing excessive stretching from occurring. Stretching factors larger than four lead to SFS (see classical papers by Dan McKenzie). Continental crust is too weak to be stretched beyond this limit. So how exactly do you propose to generate a 2000 km wide passive continental margin to start with, 5 times wider than any margin in existence? In my view this essentially falls in the land of geodynamic phantasy, so I'd like to see this justified in the paper (personally, I don't think this is possible, but I am looking forward to see the explanation you come up with!).

This is a good question. We agree with the reviewer that geodynamically it is difficult to generate 2000-km wide thinned continental crust through a single stretching event. However, such a continental crust could likely be generated via multiple instead of a single extension event, during which the underlying mantle lithosphere could temporally restabilize/regrow (McKenzie, 1978) to prevent necking and detachment as occurs in a single stretch. Since the crustal volume remains largely unchanged, its thickness may keep decreasing and length increasing. These extension events might have occurred during the ~280 Ma eruption of Panjal plume in northwestern Greater India and southern Qiangtang (Shellnutt et al., 2011; Zhai et al., 2013), the ~220 Ma breakup of Lhasa terrane next to northeastern Greater India and northwestern Australia caused by southward subduction (Zhu et al., 2011; Xu et al., 2022), the ~155 Ma eruption of plume in northeastern Greater India and northwestern Australia (Ludden, 1992; Wang et al., 2020), the ~130 Ma eruption of Comei plume in northeastern Greater India (Zhu et al., 2009), and the significant slab pull caused by double subduction during late Cretaceous (Jagoutz et al., 2015). The widely distributed extension events not only provided vertical plume push or lateral tensile force (Brune et al., 2023), but also heated and weakened the original lithosphere, which facilitate the formation of a wide stretched lithosphere (van der Pluijm & Marshak, 2004).

On the other hand, our opinion remains neutral in terms of the crustal thickness of the Greater India. From our understanding, the continental crust with a normal or thinner thickness could both be possible scenarios. The former scenario has been widely accepted by previous studies (e.g. Ingalls et al., 2016; Ding et al., 2022). The latter one has been proposed by several recent

studies (Himalandia in Liu et al. (2023); Zealandia-type in Li & Robinson, (2023)) through numerical modeling and mass-balance calculation. They think Greater India was similar to eastern Asia and southwestern Pacific nowadays, with a wide (>1000 km) and thinned continental crust as revealed by seismic probing (Gallais et al., 2019). Besides, we do not preclude the scenario that there were some seafloors spanning across Greater India induced by stretching as suggested by van Hinsbergen et al. (2017) and Yuan et al. (2021). We suggest that these would not change the major rotation features, since the stretched continental lithosphere and the new oceanic lithosphere are still more buoyant than the surrounding old oceanic lithosphere (Liu et al., 2023), thus could provide continuous resistive forces and torques altering rotation rates, similar to those discussed in the main text. In this paper, we remain neutral to these proposals and we have added related discussions in the main text (lines 351-357).

The subduction of this much continental crust also presents other problems, as is discussed, for instance, in van Hinsbergen DJ, Lippert PC, and Huang W. Unfeasible subduction? *Nature Geoscience*. 2017 Dec;10:878-9.

We have carefully looked through the related papers by van Hinsbergen, Rowley, and their colleagues (Ingalls et al., 2016; van Hinsbergen et al., 2017; Rowley & Ingalls, 2017). While van Hinsbergen et al. propose that there is a mismatch between the kinematic convergence and tectonic recovery and only the "Greater Indian Basin" model could account for the discrepancy (van Hinsbergen et al., 2017), Rowley et al. insist that a large amount of erosion of the upper crust should be considered and thus could fill the gap, and that the classic "Greater India" model still works (Ingalls et al., 2016; Rowley & Ingalls, 2017). Besides seafloor spreading and significant erosion, there are several plausible explanations for the mass discrepancy of the continental crust. First, a portion of the felsic upper crust of the normal Greater India might be dragged by the preceding slab and underwent phase change-induced densification. The negative buoyancy would make them subduct to the deeper mantle instead of exhumed to the ground (Wang et al., 2022). Second, there could also be a Greater India with a thinner continental crust as illustrated above. Thus less amount of the upper crust might be scrapped off and preserved nowadays (Liu et al., 2023; Li & Robinson, 2023). As discussed above, Greater India with a normal or stretched lithosphere are both plausible scenarios in fitting rotation features. In this paper, we remain neutral to these ideas and we have added related discussions in the main text (lines 351-357).

(2) Along the same lines, I am surprised that the recent paper by Pusok and Stegman (The convergence history of India-Eurasia records multiple subduction dynamics processes, *Sci Adv* 2020) is not cited at all. Essentially it provides some ideas for an alternative way to create a two-stage collision history, with the first collision reflecting the collision of an island arc with Eurasia followed later by a continent-continent collision. This scenario was also proposed in a number of other papers. This should at least be discussed as a potential alternative way to explain the kinematic observations.

Thank you for the reminder. We have added related discussions in the main text (lines 337-351). We have read related papers and here is our analysis. In short, the earlier "Arc-Continent

Collision" scenario is compatible with the current collision process. According to the regional plate reconstruction (Zahirovic et al., 2016), the eastern portion of the India-Eurasia convergence boundary was a single subduction zone after the Woyla arc collided with Sumatra at ca. 95 Ma (Advokaat et al., 2018). The double subduction zone is restricted to the western portion, consisting of Ladakh-Kohistan arc in the west and West Burma terrane in the east (Westerweel et al., 2019; Martin et al., 2020). It has been proposed that double subduction could exert a larger force on the subducting plate and thus could account for the high drifting rate of the Indian plate during the Cretaceous-Paleocene boundary (Jagoutz et al., 2015; Pusok et al., 2020). Considering the width of the western and the eastern convergence boundary was almost the same, there would be a larger slab pull in the west than that in the east (Fig. R5a), which would generate prominent torque resulting in extra clockwise rotation of the Indian plate. The torque and the associated extra clockwise rotation would come to an end when the island arc collided with either Indian or Eurasian continent, resulting in the extinction of the asymmetric slab pull along the boundary (Fig. R5b).

[Redacted].

But the time of the arc-continent collision is still controversial (Najman et al., 2017). Several studies propose that the arc-continent collision had been finished before 60 Ma, such as 100-80 Ma (Zaman et al., 2013), 90-80 Ma (Rehman et al., 2011), 85 Ma (Chatterjee et al., 2013; Borneman et al., 2015), and 70-60 Ma (Burg et al., 2011). If this is the case, the arc-continent collision and its associated cessation of extra clockwise plate rotation should be the overture of the Paleocene continent-continent collision. The rotation rate during the late Cretaceous, however, is heavily influenced by plume push (Extended Data Fig. 8). Thus it is difficult to isolate the signal of arc-continent collision directly.

However, some studies believe that the arc-continent collision happened at ca. 60-50 Ma

(Aitchison et al., 2007; Khan et al., 2009; Hébert et al., 2012; Bouilhol et al., 2013), which are further cited in some recent paleomagnetism and numerical modeling studies (Jagoutz et al., 2015; Westerweel et al., 2019; Martin et al., 2020; Pusok et al., 2020). If this is true, one may think that the rotation peak at 52-44 Ma might correspond to the arc-continent collision, and the peak at 33-20 Ma might reflect the continent-continent collision. But the earlier cessation of the extra clockwise rotation and the later onset of the extra counterclockwise rotation should display a continuously increasing rotation rate curve from ca. 60-50Ma to ca. 30-20 Ma (Fig. R6a), which is not consistent with the Cenozoic double peak observation (Figs. 1c, R6b).

Overall, we think the arc-continent collision before 60 Ma could be a possible scenario of the earlier collision process, which is compatible with the current Paleocene collision process.

Fig. R6. Analogue extra rotation rate of the two collision scenarios. a, Arc-continent collision at ca. 55 Ma. b, Greater India collision at ca. 55 Ma proposed in the main text.

Some detailed comments:

Line 16:

The Cenozoic India-Eurasia collision has profound impacts on the formation of the modern Tibetan plateau.

should be

The Cenozoic India-Eurasia collision has had profound impacts on the formation of the modern Tibetan plateau.

Changed. See line 16.

Line 136, and many subsequent instances:

I do not understand why in " West-bulging model", "Middle-bulging model" etc "west" and "middle" is capitalized. It should be lower-case.

We have changed the expressions throughout the main text.

Line 145:

The two modeled rotation peaks postdate the respective initial collision and complete collision by about 8 Myrs (Fig. 2a-d), which bears their formation mechanism.

should be

The two modeled rotation peaks postdate the respective initial collision and complete collision by about 8 Myrs (Fig. 2a-d), reflecting their formation mechanism.

Changed. See line 150.

Line 150: neglectable

should be

negligible

Changed. See line 154.

Line 154: This results in no noticeable rotation.

The use of the word rotation is ambiguous here, as well as through much of the rest of the paper. The issue is that all plates rotate about Euler poles at all times, ie plates are always in rotation. What you mean here is rotating about a pole located proximal to the Indian continent. You need to think about how to do that better, ie you cannot write that at some point India was not rotation, and then it started to rotate, seeing that it is ALWAYS rotating.

Thanks for the suggestion. This is a good point. We actually have also realized this. "rotation" is a more convenient word in many different circumstances throughout the manuscript, but the actual meaning might be better expressed as "azimuthal change" of a plate. We now add this definition in the abstract (line 20). And we have added the word "extra" before "rotation" in line 159 and line 203.

Line 166: Similarly, a "peak in rotation" is not the correct term here. You actually mean a peak in rotation rate. The same issue occurs in other places in the text.

Thanks again for the suggestions. We have changed the expressions throughout the main text.

Line 261: Some of the letter labels overlying "Greater India" are barely visible, especially those frequently referred to in the text like the "Saga" and "Tingri" areas.

We have redrawn Figure 4 to make it clearer.

Additional References

1. Advokaat, E. L. et al. Early Cretaceous origin of the Woyla arc (Sumatra, Indonesia) on the Australian plate. *Earth Planet. Sci. Lett.* **498**, 348-361 (2018).
2. Aitchison, J. C., Ali, J. R. & Davis, A. M. When and where did India and Asia collide? *J. Geophys. Res.* **112**, (2007).
3. Brune, S. et al. Geodynamics of continental rift initiation and evolution. *Nat. Rev. Earth Environ.* **4**, 235-253 (2023).

4. Burg, J.-P. The Asia–Kohistan–India collision: review and discussion. *Arc-continent collision* 279309 (2011).
5. Capitanio, F., Replumaz, A. & Riel, N. Reconciling subduction dynamics during Tethys closure with large-scale Asian tectonics: Insights from numerical modeling. *Geochem. Geophys. Geosyst.* **16**, 962-982 (2015).
6. Chatterjee, S., Goswami, A. & Scotese, C. R. The longest voyage: Tectonic, magmatic, and paleoclimatic evolution of the Indian plate during its northward flight from Gondwana to Asia. *Gondwana Res.* **23**, 238-267 (2013).
7. Gallais, F. et al. Crustal structure across the Lord Howe Rise, Northern Zealandia, and rifting of the eastern Gondwana margin. *J. Geophys. Res. Solid Earth* **124**, 3036-3056 (2019).
8. Hébert, R. et al. The Indus–Yarlung Zangbo ophiolites from Nanga Parbat to Namche Barwa syntaxes, southern Tibet: First synthesis of petrology, geochemistry, and geochronology with incidences on geodynamic reconstructions of Neo-Tethys. *Gondwana Res.* **22**, 377-397 (2012).
9. Khan, S. D. et al. Did the Kohistan-Ladakh island arc collide first with India? *Geol. Soc. Am. Bull.* **121**, 366-384 (2009).
10. Kirkwood, B. H., Royer, J.-Y., Chang, T. C. & Gordon, R. G. Statistical tools for estimating and combining finite rotations and their uncertainties. *Geophys. J. Int.* **137**, 408-428 (1999).
11. Li, Y. & Robinson, D. M. The India-Asia collision results from two possible pre-collisional crustal configurations of northern Greater India. *Earth Planet. Sci. Lett.* **610**, 118098 (2023).
12. Ludden, J. Radiometric age determinations for basement from Sites 765 and 766, Argo Abyssal Plain and northwestern Australian margin. Proceedings of the Ocean Drilling Program, Scientific Results. (1992).
13. Ma, Y. et al. Location of the Lhasa terrane in the Late Cretaceous and its implications for crustal deformation. *Palaeogeog. Palaeoclimatol. Palaeoecol.* **588**, 110821 (2022).
14. McKenzie, D. Some remarks on the development of sedimentary basins. *Earth Planet. Sci. Lett.* **40**, 25-32 (1978).
15. Rehman, H. U., Seno, T., Yamamoto, H. & Khan, T. Timing of collision of the Kohistan–Ladakh Arc with India and Asia: debate. *Isl. Arc* **20**, 308-328 (2011).
16. Rowley, D. & Ingalls, M. Reply to ‘unfeasible subduction?’. *Nat. Geosci.* **10**, 879-880 (2017).
17. Shellnutt, J., Bhat, G., Brookfield, M. & Jahn, B. M. No link between the Panjal Traps (Kashmir) and the Late Permian mass extinctions. *Geophys. Res. Lett.* **38**, L19308 (2011).
18. Sternai, P., Jolivet, L., Menant, A. & Gerya, T. Driving the upper plate surface deformation by slab rollback and mantle flow. *Earth Planet. Sci. Lett.* **405**, 110-118 (2014).
19. van der Pluijm, B. A. & Marshak, S. Earth structure: an introduction to structural geology and tectonics (2004).
20. van Hinsbergen, D. J. et al. Greater India Basin hypothesis and a two-stage Cenozoic collision between India and Asia. *Proc. Nat. Acad. Sci.* **109**, 7659-7664 (2012).
21. van Hinsbergen, D. J., Lippert, P. C. & Huang, W. Unfeasible subduction? *Nat. Geosci.* **10**, 878-879 (2017).
22. Wang, H.-Q. et al. The latest Jurassic protoliths of the Sangsang mafic schists in southern Tibet: Implications for the spatial extent of Greater India. *Gondwana Res.* **79**, 248-262 (2020).
23. Wang, Y., Zhang, L. & Li, Z.-H. Metamorphic densification can account for the missing felsic crust of the Greater Indian continent. *Commun. Earth Environ.* **3**, 166 (2022).
24. White, L. T. & Lister, G. S. The collision of India with Asia. *J. Geodyn.* **56**, 7-17 (2012).

25. Xu, W., Hughes, N. C., Liu, L., Zhang, W. & Liu, P. Paleogeographic reconstruction of the Paleozoic Lhasa terrane through detrital zircon mixing modeling. *Geophys. Res. Lett.* **49**, e2022GL100160 (2022).
26. Yi, Z. et al. A quasi-linear structure of the southern margin of Eurasia prior to the India-Asia collision: First paleomagnetic constraints from Upper Cretaceous volcanic rocks near the western syntaxis of Tibet. *Tectonics* **34**, 1431-1451 (2015).
27. Zaman, H., Otofujii, Y.-i., Khan, S. R. & Ahmad, M. N. New paleomagnetic results from the northern margin of the Kohistan Island Arc: Un-ending thermo-tectonic activities in the India-Asia collision zone. *Arab. J. Geosci.* **6**, 1041-1054 (2013).
28. Zhai, Q.-G. et al. SHRIMP zircon U-Pb geochronology, geochemistry and Sr-Nd-Hf isotopic compositions of a mafic dyke swarm in the Qiangtang terrane, northern Tibet and geodynamic implications. *Lithos* **174**, 28-43 (2013).
29. Zhu, D.-C. et al. The Lhasa Terrane: Record of a microcontinent and its histories of drift and growth. *Earth Planet. Sci. Lett.* **301**, 241-255 (2011).

REVIEWERS' COMMENTS

Reviewer #1 (Remarks to the Author):

The authors have done quite an effort to revise the paper, and all my comments have been addressed. Accordingly, I have only a few small comments - partly things I had overlooked in the first round, partly new issues that came with the revisions.

On lines 94-98 you write now that the first peak is not as robust and that this is likely due to the discrepancy in the adopted absolute reference frames. I suggest that, like you explain in the rebuttal letter, you can, for "evidence", add a sentence that, if you replace the slab-fitting reference frame by a hotspot reference frame, you still get a 52-44 Ma peak.

Another issue is about the shift in the center of mass, which is newly introduced. Is this just something that you prescribe (and which is the same in Figures 2 and Extended Data Figures 5a and 6, and not considered in Extended data Figures 5c and 7) or do you actually compute it for all your models? You explain why it should move westward, but shouldn't it also move southward relative to the continent, because material is lost in the north? Or do you only consider the westward component of motion? These questions apply to both your numerical models and your simplified quantitative calculation.

Related to this, you refer to Extended Data Figure 5d on line 205. You explain in your rebuttal letter and the methods section that the east-bulging case is better combined with no motion of the center of mass, but here it does not become clear why you refer to panel d rather than b.

Caption Figure 3: I think you should add to the figure caption, like you did in your rebuttal letter, that the dashed lines outline the subducted portion.

line 345: You mean, a continuously increasing counterclockwise rotation rate, i.e. that the rotation rate was always counterclockwise, despite the additional clockwise rotation?

lines 531-533: I would rewrite this sentence as follows: "... which has an inherent azimuth variation component, except in special circumstances, such as when the Euler pole is located at the North/South pole, or at the equator 90° of longitude away from the central reference point"

lines 570-572: You write in your rebuttal letter that temperature is not considered in the continental lithosphere, but you don't write it explicitly here. If that is so, I think you should also write it here for clarity.

Extended data Table 3: You added eta_cut-off, but I think you need some explanation that these are the upper and lower limits of viscosity considered. Is viscosity given in units of 10^{21} Pas? In general, I find the mix of dimensionless and dimensional numbers a bit confusing. You need to be clear about what is what - for dimensional numbers give the units, and for dimensionless ones give the normalizing (or reference) value.

Minor comments:

line 146: perhaps better "timing" rather than "tempo"?

line 204: add "simplified" before "quantitative calculation"?

line 220: better "abnormal"?

line 635: remove "peak"?

line 639: instead of "is larger" perhaps better "reaches a larger value", since it is not always larger?

Reviewer #2 (Remarks to the Author):

The authors have done an excellent job in revising the paper, addressing all reviewer comments in detail. I am happy for the paper to be published in its present form.

Reviewer #1 (Remarks to the Author):

The authors have done quite an effort to revise the paper, and all my comments have been addressed. Accordingly, I have only a few small comments - partly things I had overlooked in the first round, partly new issues that came with the revisions.

Thank you very much for the supportive comments and valuable suggestions that have greatly improved our manuscript. The remaining issues have been corrected now.

On lines 94-98 you write now that the first peak is not as robust and that this is likely due to the discrepancy in the adopted absolute reference frames. I suggest that, like you explain in the rebuttal letter, you can, for "evidence", add a sentence that, if you replace the slab-fitting reference frame by a hotspot reference frame, you still get a 52-44 Ma peak.

Thank you for your suggestion. We have added the related sentence in the main text. See lines 102-104 in inline markup mode.

Another issue is about the shift in the center of mass, which is newly introduced. Is this just something that you prescribe (and which is the same in Figures 2 and Extended Data Figures 5a and 6, and not considered in Extended data Figures 5c and 7) or do you actually compute it for all your models? You explain why it should move westward, but shouldn't it also move southward relative to the continent, because material is lost in the north? Or do you only consider the westward component of motion? These questions apply to both your numerical models and your simplified quantitative calculation.

Thanks for pointing this out. Since the exact movement of mass center in the models is difficult to extract, what's presented is a simplified version of the movement of mass center corresponding to the simplified torque analysis. As the numerical models are designed to be free-slip and without any prescribed forces on the boundaries, the rotation of the subducting plate and the shift of the center of mass are both natural outcomes of the model without additional prescription. In order to explain the observed rotation feature of the numerical models, we introduced the movement of the center of mass in the simplified torque calculations, a phenomenon which is observed in the west- and middle-bulging models accompanying significant counterclockwise rotations. As for the east-bulging model, the magnitude of counterclockwise rotation and the offset of the center of mass is relatively minor, therefore the torque calculations with a fixed mass center should be more appropriate.

We agree with the reviewer that the center of mass should move southward with the collision going on. We have redrawn the figures to fix this problem (Fig. 2, Supplementary Figs. 6,7). Actually, these changes do not influence the original explanation because the length of the lever arm is unaltered (Fig. 2). Therefore the southward movement of the center of mass does not change the resistive torques in the simplified calculations. We have added a related explanation in Methods. See lines 474-476.

Related to this, you refer to Extended Data Figure 5d on line 205. You explain in your rebuttal letter and the methods section that the east-bulging case is better combined with no motion of the center of mass, but here it does not become clear why you refer to panel d rather than b.

Thanks for pointing this out. We have made some clarifications in the figure caption that can guide the readers to understand the different choices of torque analysis for different models by referring to the Methods part. See figure captions below Supplementary Figure 5 in the separate supplementary file. We avoid to discuss the movement of mass center in the main text to keep the main text concise and to avoid redundancy, since all related discussions are compactly shown in the Methods section.

Caption Figure 3: I think you should add to the figure caption, like you did in your rebuttal letter, that the dashed lines outline the subducted portion.

Thank you for your suggestion. We have added the related sentence in the caption of Figure 3. See lines 222-223.

line 345: You mean, a continuously increasing counterclockwise rotation rate, i.e. that the rotation rate was always counterclockwise, despite the additional clockwise rotation?

Here we use the expression "a continuously increasing rotation rate" to describe the hypothetical variation in the rotation rate of the Indian plate, i.e. the earlier weakening of the clockwise rotation induced by arc-continent collision, and the latter strengthening of counterclockwise rotation induced by continent-continent collision. There could be a period of negative (i.e. clockwise) rotation rate when there was a double-subduction system exerting large enough extra clockwise torque on the Indian plate. But the observed Cenozoic rotation rate of the Indian plate was positive (i.e. counterclockwise) almost all the time (Fig. 1c, Supplementary Fig. 2), and the double-peak feature also seems inconsistent with the predictions. Therefore the scenario in which the arc-continent happened at ca. 60-50 Ma is not favored.

We now change this expression to "a continuously increasing rotation rate (assuming negative for clockwise rotation and positive for counterclockwise rotation)". See lines 353-354.

lines 531-533: I would rewrite this sentence as follows: "... which has an inherent azimuth variation component, except in special circumstances, such as when the Euler pole is located at the North/South pole, or at the equator 90° of longitude away from the central reference point"

Thank you for your suggestion. We have changed the expression here. See lines 387-389.

lines 570-572: You write in your rebuttal letter that temperature is not considered in the continental lithosphere, but you don't write it explicitly here. If that is so, I think you should also write it here for clarity.

Thank you for your suggestion. We have added related description in the Methods. See line 429.

Extended data Table 3: You added eta_cut-off, but I think you need some explanation that these are the upper and lower limits of viscosity considered. Is viscosity given in units of 10^{21} Pas? In general, I find the mix of dimensionless and dimensional numbers a bit confusing. You need to be clear about what is what - for dimensional numbers give the units, and for dimensionless ones give the normalizing (or reference) value.

Thank you for your suggestion. We have added related descriptions below the Supplementary Table 3. See table caption below the Supplementary Table 3. We have also converted all non-dimensional values to dimensional values, including those in Supplementary Tables 2 and 3.

Minor comments:

line 146: perhaps better "timing" rather than "tempo"?

line 204: add "simplified" before "quantitative calculation"?

line 220: better "abnormal"?

line 635: remove "peak"?

line 639: instead of "is larger" perhaps better "reaches a larger value", since it is not always larger?

Thank you again for your suggestions. We have changed related expressions in the main text and Methods. See lines 152, 209, 227, 494, 499.

Reviewer #2 (Remarks to the Author):

The authors have done an excellent job in revising the paper, addressing all reviewer comments in detail. I am happy for the paper to be published in its present form.

We appreciate your thorough review, which has greatly improved our manuscript.